# Earwig Releases Provide Accumulative Biological Control of the Woolly Apple Aphid over the Years

**DOI:** 10.3390/insects14110890

**Published:** 2023-11-18

**Authors:** Georgina Alins, Jaume Lordan, Neus Rodríguez-Gasol, Judit Arnó, Ainara Peñalver-Cruz

**Affiliations:** 1Fruit Production Program, Institut de Recerca i Tecnologia Agroalimentàries (IRTA), Parc Agrobiotech Lleida, Parc de Gardeny, Fruitcentre Building, 25003 Lleida, Spain; jaume.lordan@irta.cat (J.L.); neus.rodriguez.gasol@slu.se (N.R.-G.); 2Department of Ecology, Sveriges lantbruksuniversitet (SLU), P.O. Box 7044, 75007 Uppsala, Sweden; 3Sustainable Plant Protection Program, Institut de Recerca i Tecnologia Agroalimentàries (IRTA), Ctra de Cabrils km.2, 08348 Cabrils, Spain; judit.arno@irta.cat; 4Sustainable Plant Protection Program, Institut de Recerca i Tecnologia Agroalimentàries (IRTA), Av. Alcalde Rovira Roure, 191, 25198 Lleida, Spain; ainara.penalver@irta.cat

**Keywords:** *Forficula auricularia*, *Eriosoma lanigerum*, *Aphelinus mali*, natural enemies, pest management, nature-based solutions, orchard

## Abstract

**Simple Summary:**

The woolly apple aphid (Eriosoma lanigerum) is a worldwide pest that causes damage to shoots, branches, trunks, and roots and reduces the capacity of apple trees to sprout and bloom. Even though this pest has a diverse guild of natural enemies, the parasitoid Aphelinus mali and the European earwig (Forficula auricularia) are the ones that have the most potential to regulate this pest. In our study we assessed the impact of earwig releases over time on woolly apple aphid populations. To do this, we released earwigs once per season in consecutive years and we compared woolly apple aphid abundance with respect to a control treatment (without earwig release). Overall, we found that earwig releases reduced the size of the colonies, but this effect was only evident from the second year onwards. These results highlighted the importance of considering time on augmentative biological control strategies.

**Abstract:**

Nature-based solutions, such as biological control, can strongly contribute to reducing the use of plant protection products. In our study, we assessed the effect of augmentative releases of the European earwig (*Forficula auricularia*) to control the woolly apple aphid (*Eriosoma lanigerum*), a worldwide pest that causes serious damage to apple trees. The trials were carried out in two organic apple orchards located in Catalonia (NE Spain) from 2017 to 2020. Two treatments were compared: with vs. without earwig release. For the treatment, 30 earwigs per tree were released by means of a corrugated cardboard shelter. These releases were performed once per season and were repeated every year. We periodically assessed the length of the woolly apple aphid colonies, the number of colonies per tree, the percentage of aphids parasitized by *Aphelinus mali*, and the number of earwigs per shelter. Our results showed that earwig releases reduced the length of the colonies, but this effect was noticeable only for the second year onwards. Moreover, we found that those releases were compatible with *A. mali*. Overall, we demonstrated the positive impact of earwig releases on the woolly apple aphid control and the importance of considering time on augmentative biological control strategies.

## 1. Introduction

The reduction in the use of pesticides is a social demand that is reflected in policies. For instance, the European Commission released the “European Green Deal” in 2019, which mainly aims “to transform the EU into a fair and prosperous society, with a modern, resource-efficient and competitive economy”. As a result, it developed the “Farm to Fork Strategy”, which urges a reduction in the use of pesticides of around 50% by 2030. To achieve this general goal, useful alternative pest management strategies must be developed, adapted to the local agroecological context, and finally implemented by farmers. In this regard, nature-based solutions such as biological control can strongly contribute to increasing the environmental sustainability of farms [1,2]. However, even though many successful examples of biological control have been reported around the world [3], their practical use is limited [2]. Each of the biological control strategies (classical biological control, augmentative biological control, and conservation biological control) shows different levels of efficacy according to the combination of several factors, such as pest species, crop, guild of natural enemies, and climate among others. Classical biological control consists of an intentional introduction of a foreign natural enemy for permanent establishment and long-term pest control. In augmentative biological control, natural enemies are mass-reared and released on a scheduled basis. Furthermore, this strategy has two modalities (inoculative and inundation biological control) that differ based on the population that controls the pest: in inoculative biological control, it is the progeny that effectuate the control, whereas in inundation biological control, it is the released population. In addition, when a population of natural enemies is inoculated, pest control for several seasons is expected, whereas in inundation biological control, only an immediate effect is expected. Conservation biological control is a set of practices to protect and boost natural enemies’ populations to increase pest control. In any case, the main objective of biological control is to reduce pest populations to lessen economic losses rather than to eliminate the pest. Maintaining a non-damaging level of pests can contribute to sustaining a guild of natural enemies to manage pests in the long term; otherwise, natural enemies may abandon the farm in search for food and then, pest get out of control again [4,5,6].

Unlike pesticides, actions to increase the pest self-regulation capacity of an agroecosystem often have slow responses and are not efficient enough [7,8,9]. Since pests must be managed to minimize current crop losses and to maintain or decrease pest population for the next season, biological control could be a good tool to low down the use of insecticides [10,11].

In this article, we describe the effect of a predator release on the control of an aphid pest population over time. We chose a Mediterranean apple (*Malus domestica* (Borkhausen)) orchard as multiannual agroecosystem model, the woolly apple aphid (*Eriosoma lanigerum* (Hausmann) (Hemiptera: Aphididae)) as the exotic pest, and earwigs (*Forficula auricularia* L. (Dermaptera: Forficulidae)) as a native and generalist predator. Even though this aphid is native to North America, it is present in all apple-growing areas and causes damage to shoots, branches, trunks, and roots. When colonies develop on shoots and branches, the capacity of the apple trees to sprout and bloom is drastically reduced, as it destroys the buds, diminishing the productive potential of the trees [12]. In addition, the woolly apple aphid excretes honeydew and—in case of severe infestation—aesthetically damages the fruits. Therefore, the economic value of the yield is reduced [13].

As most of the aphids, the woolly apple aphid has associated a diverse guild of natural enemies: the parasitoid *Aphelinus mali* (Haldeman) (Hymenoptera: Aphelinidae), which is specific to the woolly apple aphid, and predators such as spiders (Araneae), ladybird beetles (Coleoptera: Coccinellidae), lacewings (Neuroptera: Chrysopidae), hoverflies (Diptera: Syrphidae), and earwigs [14,15,16]. From these natural enemies, *A. mali* and earwigs are the most cited and the ones that have more potential to regulate this pest [17,18,19,20,21]. Earwigs in particular have been reported as important predators of other pests like pysllids [22], scales [23], flies [24,25], mites [26], and aphids [19,20,27,28,29,30,31]. They are omnivorous insects that, apart from arthropods, also feed on fungi, lichens, and plant material [32,33]. Therefore, earwigs are good candidates to be used in augmentative biological control, since in the absence or low densities of pests, the orchard will always offer alternative food resources. Regardless of the above and to the best of our knowledge, only four papers published in Science Citation Index journals deal with augmentative releases of earwigs for pest control, and they indicate contradictory results. For instance, Carroll et al. [27] reported that earwig releases in apple orchards successfully controlled the green apple aphid (*Aphis pomi* De Geer (Hemiptera: Aphididae)), but one year later, the same group of research published opposite results [29]. Furthermore, Dib et al. [34] found that earwig release did not reduce rosy apple aphid (*Dysaphis plantaginea* Passerini (Hemiptera: Aphididae)) populations. Regarding the woolly apple aphid, earwig releases were ineffective in controlling this pest according to Carroll et al. [29] but satisfactory according to Orpet et al. [35]. Moreover, as far as we know, there are not studies that analyse the contribution of augmentative release of earwigs over time, as the reported studies above were conducted for one year.

Given the above, the aim of this study is to assess the impact of augmentative releases of earwigs over time on woolly apple aphid populations. We hypothesise that the release of earwigs before peak abundance will prevent outbreaks. We also explore the effects of those releases on parasitism of the woolly apple aphid colonies.

## 2. Materials and Methods

### 2.1. Study Site

Trials were carried out in two mature organic apple orchards located in the fruit tree-growing area of Lleida (Catalonia, NE Spain): from 2017 to 2019 in Anglesola (Orchard 1, 41°39′25.1″ N 1°03′24.1″ E) and from 2019 to 2020 in El Palau d’Anglesola, (Orchard 2, 41°39′34.0″ N 0°51′23.9″ E). Orchard 1 is 0.5 ha, and it was planted in 1998 with “Fuji” grafted onto M9 (4 × 1.3 m spacing). Orchard 2 is 1.5 ha, and it was planted in 2000 also with “Fuji” grafted onto M9 (4 × 1.25 m spacing). Both orchards were sprayed with plant protection products allowed by the EU organic growing regulations: azadirachtin to control the rosy apple aphid (*Dysaphis plantaginea* Passerini (Hemiptera: Aphididae)), granulosis virus to control the codling moth (*Cydia pomonella* L. (Lepidoptera: Tortricidae)), and lime sulphur to control apple scab (*Venturia inaequalis* Cooke). In addition, the tree rows of Orchard 2 were covered with nets (from June to October in 2019 and from May to October in 2020) to protect apples from codling moth.

### 2.2. Treatments

Two treatments were compared (earwig release and control) in a completely randomized design with 10 replicates per field, in which each replicate was formed by one tree. The first treatment consisted of the release of 30 earwigs per tree. The release was performed every year at the beginning of the experiment (6 June 2017, 8 June 2018 and 7 May 2019 in Orchard 1 and 8 May 2019 and 14 May 2020 in Orchard 2) by means of a shelter. The shelter constituted a corrugated cardboard cylinder (9 cm diameter × 12 cm height) inserted into a PVC tube [36], and it was placed horizontally next to a woolly apple aphid colony that was used for the surveys (Figure 1A). The control treatment did not receive any artificial shelter, and no earwigs were released. The minimum distance between replicates was 8 m to avoid interference between them. According to Moerkens et al. [37], when earwigs of a single-brood population are released, they remain within a radius of 8.7 m, and those from double-brood populations remain within a radius of 2.3 m. In our area, both populations co-exist [36], therefore 8 m is a compromise distance that allows the performance of the trial in each orchard.

Previously to the treatment randomization, the level of woolly apple aphid infestation was visually assessed in all the trees of the orchard according to a categorical scale (0: 0 colonies per tree, 1: 1–25 colonies per tree, 2: 26–100 colonies per tree, 3: >100 colonies per tree). When possible, only trees in Category 2 were used to conduct the experiment. In each of those trees, one shoot per tree infested by the woolly apple aphid was selected to be evaluated.

The treatments were randomized every year, and the trees sampled during the previous year were not used in the following year.

### 2.3. Assessments

At the beginning of the trial (first week, when earwigs were released), the length of the woolly apple aphid colonies present in the selected shoots and the number of woolly apple aphid colonies per tree were assessed (Figure 1B). In addition, the presence of parasitized aphids by *A. mali* was visually estimated according to a categorical scale [21]. The length of the woolly apple aphid colonies and the abundance of parasitized aphids were assessed weekly, whereas the number of woolly apple aphid colonies and the number of earwigs per shelter were recorded fortnightly. The spontaneous populations of earwigs before the releases were not assessed, and the first sampling was performed in the third week of the trial. Once the earwigs were counted, they were immediately released into the same shelter. The difference in sampling frequencies between those natural enemies was set to reduce earwig manipulation.

### 2.4. Statistical Analysis

The length of the woolly apple aphid colonies, the percentage of parasitized colonies, the number of woolly aphid colonies per tree, and the number of earwigs per shelter were analysed per each sampling date and orchard. In addition, to analyse the effect of the treatments on the length of the woolly apple aphid colonies and on the number of colonies per tree along the sampling period, it was calculated the area under the curve of those variables. To do this, it was assumed that the increase or decrease between two consecutive sampling dates was linear. We used Formula (a), and the units of these new variables were expressed as colony length·day and number of colonies per tree·day, respectively.

(a) $A=\sum_{i=1}^{n}\left(d\times y_{i}+d\times \frac{\left|y_{i}-y_{i+1}\right|}{2}\right)$, where *A* is the area under the curve; *i* and *n* are the first and last sampling data, respectively; and *d* is the number of days between samplings (7 in the case of colony length and 14 in the case of number of colonies per tree).

When data did not meet ANOVA assumptions, non-parametric test were used (Welch or Kruskal–Wallis tests). A significance level of *p* ≤ 0.05 was considered for all the analyses. Data were analysed using the JMP statistical software package (Version 16.0.0; SAS Institute, Inc., Cary, NC, USA).

## 3. Results

The spontaneous presence of woolly apple aphid differed between years and orchards (Figure 2 and Figure 3). In Orchard 1, the length of the colonies decreased over years, with peaks of 19.2 ± 5.1 cm in 2017, 8.3 ± 1.4 cm in 2018, and 5.3 ± 0.8 in 2019. In contrast, the number of colonies per tree increased over years, with maximums of 21.2 ± 5.4 colonies per tree in 2017, 99.9 ± 5.4 colonies per tree in 2018, and 357.7 ± 41.6 colonies per tree in 2019. In Orchard 2, the length of the colonies was similar between years, with peaks of 2.0 ± 0.5 cm in 2019 and 2.6 ± 0.3 in 2020. Additionally, contrary to what we observed in Orchard 1, the number of colonies per tree decreased over the years, with maximums of 274.2 ± 24.7 colonies per tree in 2019 and of 137.0 ± 15.0 colonies per tree in 2020.

Regarding the effect of the earwig release on woolly apple aphid populations, in both orchards and in the first year, a release of 30 earwigs per tree did not cause a statistically significant change in the level of woolly apple infestation in either colony length (Figure 2A,D, Table 1) or the number of colonies per tree (Figure 3A,D and Table 2). In the second year, the woolly apple aphid colony length was statistically shorter in the release treatment compared to the control in both orchards (Figure 2B,E and Table 1). These results were also observed in the third year of release in Orchard 1 (Figure 2C and Table 1). However, in general, there were no statistically significant differences in the number of colonies per tree (Figure 3C and Table 2).

Concerning earwigs, the number of earwigs per shelter was lower than the number of earwigs released at the beginning of the trial in both orchards and in all the samplings performed during the first and second years of releases (Figure 4). Within orchards, the number of earwigs per shelter was not statistically different between the first and second years of releases. In contrast, in the third year, the number of earwigs per shelter in the first assessment (two weeks after release) exceeded the number of earwigs released (Figure 4A), and the number of earwigs per shelter was statistically higher than in the first and second years of releases.

No significant differences were found in the percentage of the colony parasitized by *A. mali* for most of the sampling dates (Table 3).

## 4. Discussion

Our results show for the first time the effect of a release of earwigs conducted in spring and repeated in the following years on biological control of the woolly apple aphid and the timing needed to reduce pest abundance. In this regard, we demonstrated that earwig releases contribute to biological control of the woolly apple aphid and that this effect is not noticed in the first year of release but from the second onwards. Orpet et al. [35] performed a similar trial, but earwigs were released several times throughout spring and summer for only one year. This study reported that the reduction in woolly apple aphid abundance was achieved in the same year of releases. The difference in timing is probably due to a higher number of earwigs released per tree (between 47–230 vs. 30) and a much lower abundance of woolly apple aphid (maximum of an average of about 3 colonies per tree) compared to our orchards (21.2–357.7 colonies per tree). Therefore, we suggest that the difference between Orpet et al.’s [35] and our results could be due to the high difference in the ratio of earwigs released/woolly apple aphid colonies per tree: 15.6–76.6 vs. 0.1–1.4. In addition, it should be noticed that earwigs are omnivorous [32,33], so in the first year of releases, the effect of an increase in earwig population on woolly apple aphid abundance could be diluted by the lack of limitation of food resources (other aphids and soft body arthropods, lichens, fungi, etc.) present in the mature orchards.

In the conditions of the present study, the earwigs released in the first and second year probably caused an accumulative increase in the resident earwig population in the second and third years, respectively. Moreover, the unlimited food in the orchards could support these raised populations over the years. Earwigs mate during summer and lay eggs only once in summer and autumn (in the case of single-brood populations), or twice in summer–autumn and winter–early spring (in case of double-brood populations) [38,39]. So, we hypothesize that the effective boosting of earwig populations regarding biological control was due to combined action of the offspring of the earwigs released in the previous year with the earwigs released in the present year.

As we found that earwig release reduced the colony length in the second year, we expected to catch more earwigs than in the first one. However, the number of earwigs recorded was similar between those years and within orchards. This fact could be explained by the medium efficacy of cardboard shelters to accurately assess earwig populations. Shaw et al. [40] stated that cardboard shelters are more effective in young trees rather than adult trees because of the higher abundance of natural shelters in the older ones. In addition, Moerkens et al. [39] reported an increase in earwigs in shelters after the pear harvest because pears are clustered, providing natural refuges for earwigs. Therefore, we suggest that the increase in earwig population in the second year was not detected by the cardboard shelters, as the number of natural shelters in those mature orchards was high enough. In contrast, in the third year of release, we recorded more earwigs than in the previous years, probably due to the fact that the number of natural shelters were not enough to provide refuge to the accumulated population, and therefore, earwigs occupied the artificial shelters. Overall, we suggest that the number of earwigs per shelter is not an accurate method of determining the abundance of earwig populations. Therefore, a mathematical model for accurately predicting earwig populations should be developed. This model could be based on factors that are known to estimate or affect earwig populations: the output of one or a combination of different sampling methods (corrugated cardboard shelters, frass assessment [41], or video recording [42]); local factors, like ground cover management [41,43] or the abundance of hiding places [36,39]; landscape factors, such as adjacent habitats [44]; and other variables.

On the other hand, the fact that a reduction in the colony length was only recorded in the release treatment confirms the low rate of dispersal of earwigs, also reported by Moerkens et al. [37]. Therefore, the upscaling of earwig releases to manage the woolly apple aphid requires more research to find a balance between the number of points of release per orchard and its costs.

Even though earwig releases reduced the length of the colonies from the second year onwards, this reduction did not eliminate the woolly apple aphid colonies, and therefore, the releases did not diminish the number of colonies per tree. These results could be explained by the ratio of earwigs released/woolly apple aphid colonies per tree. If the number of earwigs released had been higher (>30 earwigs per tree), we could have found a greater reduction in the woolly apple aphid populations. Therefore, more research should be carried out to find a compromise between released earwig efficacy and feasibility. In addition, it is unknown whether a huge release of earwigs could cause an ecological imbalance and a promotion of intraspecific competition for food resources. In our study, we found that a release of 30 earwigs per tree and season is compatible with *A. mali* since the percentage of parasitized woolly apple aphids was similar between treatments in most of the years and orchards. Several authors reported the compatibility and even the complementarity of spontaneous populations of earwigs and *A. mali* [18,45] and also earwigs with other natural enemies, like Coccinellid larvae [42].

## 5. Conclusions

This study shows the positive impact of earwig releases on woolly apple aphid control over time in a warm apple-growing area in which the abundance of this pest is noticeable. A release of 30 earwigs per tree in spring reduced the length of the colonies, but this effect is only evident from the second year onwards. These results highlight the importance of considering time on augmentative biological control strategies. On the other hand, the effects of multiple releases throughout the season should be tested in the context of high populations of this pest to determine if biological control can be achieved in the first year of releases. More research should be performed to find an accurate model for estimating earwig populations in orchards that will allow the prediction of earwig predation on woolly apple aphid colonies and the optimal number of release points per orchard needed to manage this pest.

## Figures and Tables

**Figure 1 insects-14-00890-f001:**
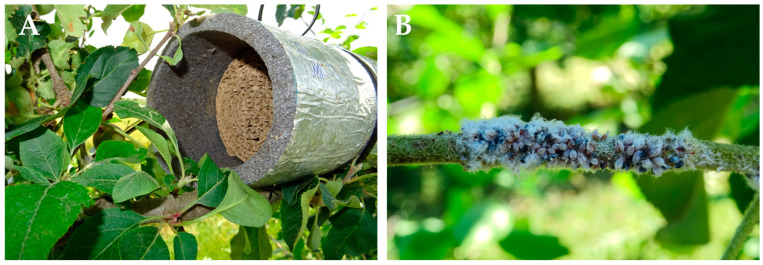
(**A**): Detail of the shelter used to release earwigs. (**B**): Woolly apple aphid colony with some parasitized aphids.

**Figure 2 insects-14-00890-f002:**
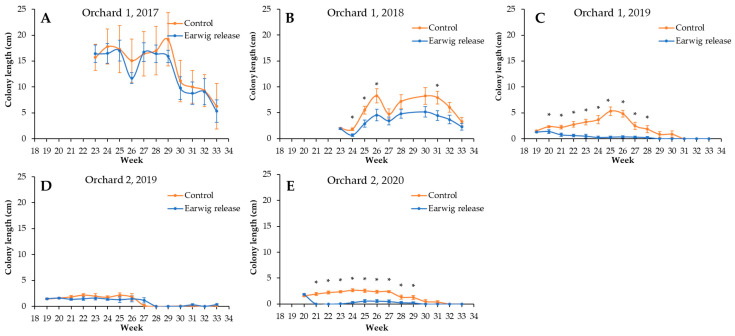
Mean woolly apple aphid colony length (cm) per week, orchard, and year. Vertical bars show standard error. * = *p* <0.05. (**A**): Orchard 1, year 2017. (**B**): Orchard 1, year 2018. (**C**): Orchard 1, year 2019. (**D**): Orchard 2, year 2019. (**E**): Orchard 2, year 2020.

**Figure 3 insects-14-00890-f003:**
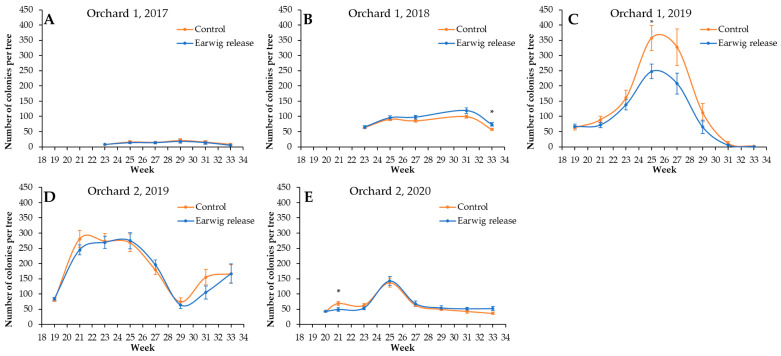
Mean number of woolly apple aphid colonies per tree, week, orchard, and year. Vertical bars show standard error. * = *p* <0.05. (**A**): Orchard 1, year 2017. (**B**): Orchard 1, year 2018. (**C**): Orchard 1, year 2019. (**D**): Orchard 2, year 2019. (**E**): Orchard 2, year 2020.

**Figure 4 insects-14-00890-f004:**
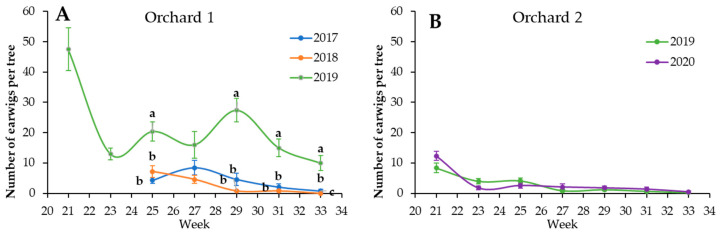
Mean number of earwigs per shelter, week, and year within orchard. Vertical bars show standard error. Different letters indicate significant differences among years (*p* < 0.05). Absence of letter indicates no statistically significant differences among years. (**A**): Orchard 1. (**B**): Orchard 2.

**Table 1 insects-14-00890-t001:** Area under the line of woolly apple aphid colony length (mean ± standard error).

Orchard	Year	Control	Earwig Release	DF	F	*p*
Orchard 1	2017	942.5 ± 259.8	1091.2 ± 115.6	1,15	0.3211	0.5799
Orchard 1	2018	381.8 ± 52.6	231.7 ± 41.5	1,19	5.0138	**0.0380**
Orchard 1	2019	276.2 ± 36.7	52.9 ± 15.3	1,19	31.544	**<0.0001**
Orchard 2	2019	126.1 ± 21.6	111.4 ± 21.2	1,18	0.2345	0.6344
Orchard 2	2020	170.4 ± 16.6	41.4 ± 12.1	1,18	40.4831	**<0.0001**

Significant differences between treatments are shown in bold.

**Table 2 insects-14-00890-t002:** Area under the line of number of woolly apple aphid colonies per tree (mean ± standard error).

Orchard	Year	Control	Earwig Release	DF	F/χ^2^	*p*
Orchard 1	2017	1165.5 ± 312.1	1285.2 ± 294.8	1,19	0.0777	0.7836
Orchard 1	2018	4555.6 ± 273.1	4122.3 ± 119.6	1,19	2.1124	0.1633
Orchard 1	2019	14,515.2 ± 1672.4	20,692.7 ± 2924.9	1	2.0629	0.1509
Orchard 2	2019	21,261.3 ± 1503.1	22,638 ± 1797.6	1,18	0.3365	0.5695
Orchard 2	2020	7700 ± 725	7736.1 ± 648.3	1,19	0.0014	0.9708

**Table 3 insects-14-00890-t003:** Mean percentage of woolly apple aphid colony parasitized per week, orchard, and year (mean ± standard error).

Orchard	Year	Week	Control	Earwig Release	DF	F/χ^2^	*p*
Orchard 1	2017	23	24.0 ± 6.4	22.5 ± 3.8	1,19	0.0401	0.8436
		24	23.0 ± 8.5	28.0 ± 7.9	1,19	0.1856	0.6717
		25	15.0 ± 4.1	23.0 ± 8.5	1	0.1206	0.7283
		26	35.5 ± 8.7	62.0 ± 8.0	1,19	5.0230	**0.0379**
		27	34.5 ± 10.4	40.0 ± 10.3	1,19	0.1416	0.7111
		28	32 ± 12.4	32.0 ± 12.4	1,19	0.0000	1.0000
		29	30.6 ± 8.4	33.0 ± 11.0	1,18	0.0302	0.8641
		30	65.0 ± 12.0	83.3 ± 7.7	1,14	1.8291	0.1993
		31	78.3 ± 5.3	84.3 ± 5.1	1,12	0.6611	0.4334
		32	91.7 ± 4.4	95.7 ± 0.7	1,12	0.9627	0.3476
		33	98.0 ± 1.2	96.7 ± 1.1	1,10	0.6890	0.4280
	2018	23	0.0 ± 0.0	0.0 ± 0.0	-	-	-
		24	9.5 ± 6.8	11.7 ± 5.9	1,15	0.0478	0.8301
		25	24.0 ± 6.4	15.5 ± 4.9	1,19	1.1092	0.3062
		26	21.5 ± 6.7	7.5 ± 3.8	1,19	3.3189	3.3189
		27	43.5 ± 7.6	30.5 ± 9.5	1,19	1.1415	0.2995
		28	32.5 ± 7.2	21.5 ± 6.7	1,19	1.2568	0.2770
		30	59.0 ± 8.5	37.0 ± 9.9	1,19	2.8564	0.1083
		31	70.5 ± 9.4	77.0 ± 8.5	1,19	0.2644	0.6134
		32	95.0 ± 0.0	92.2 ± 2.8	1	1.1111	0.2918
		33	95.0 ± 0.0	91.9 ± 3.1	1	1.0000	0.3173
	2019	19	6.5 ± 2.0	2.5 ± 1.1	1,19	3.0968	0.0954
		20	23.5 ± 5.1	28.6 ± 9.3	1,16	0.2653	0.6140
		21	38.0 ± 10.7	55.7 ± 16.2	1,16	0.9065	0.3561
		22	13.9 ± 4.1	17.5 ± 14.2	1,12	0.1084	0.7481
		23	20.0 ± 5.1	7.5 ± 2.5	1,10	1.2355	0.2951
		24	13.5 ± 5.0	51.3 ± 48.8	1	0.1886	0.6641
		25	15.5 ± 4.2	28.3 ± 16.4	1,12	1.3109	0.2765
		26	14.0 ± 3.2	51.3 ± 19.4	1,13	8.7151	**0.0121**
		27	54.0 ± 11.5	47.5 ± 4.8	1	0.1849	0.6672
		28	68.6 ± 12.4	75.0 ± 25.0	1,8	0.0582	0.8163
		29	66.0 ± 14.7	100.0	1,5	0.8920	0.3984
		30	56.7 ± 21.9	50.0	1,3	0.0233	0.8928
		31	100.0 ± 0.0	100.0	-	-	-
Orchard 2	2019	19	10.5 ± 2.5	13.3 ± 5.2	1,18	0.2563	0.6192
		20	28.0 ± 4.6	28.3 ± 7.9	1,18	0.0014	0.9707
		21	17.3 ± 4.3	22.2 ± 5.3	1,18	0.5401	0.4724
		22	25.5 ± 5.3	19.3 ± 4.8	1,16	0.6730	0.4248
		23	31.0 ± 5.8	28.6 ± 6.6	1,16	0.0755	0.7873
		24	49.0 ± 4.3	30.6 ± 8.3	1	2.7056	0.1000
		25	45.0 ± 7.9	61.7 ± 11.7	1,15	1.4983	0.2411
		26	79.5 ± 6.3	66.4 ± 10.6	1,16	1.2857	0.2746
		27	97.9 ± 1.5	81.0 ± 6.4	1,11	9.0793	**0.0130**
		28	95.0	98.8 ± 1.3	1,4	1.8000	0.2722
		29	-	100.0	-	-	-
		30	99.0		-	-	-
		31	95.0 ± 5.0	82.5 ± 2.5	1,3	5.0000	0.1548
		32	95.0	100.0 ± 0.0	-	-	-
		33	30.0	75.0 ± 7.6	1,3	8.6786	0.0985
	2020	20	15.0 ± 2.1	17.5 ± 2.4	1,19	0.6164	0.4426
		21	29.0 ± 5.5	100.0 ± 0.0	1	15.2226	**<0.0001**
		22	42.5 ± 5.5	100.0	1,10	9.9728	**0.0116**
		23	43.3 ± 7.3	50.0 ± 50.0	1,1.04	0.0174	0.1319
		24	45.6 ± 6.3	37.5 ± 22.5	1,3.4833	0.1188	0.3447
		25	35.0 ± 7.0	52.5 ± 19.3	1,12	1.1698	0.3026
		26	53.3 ± 8.0	91.7 ± 1.7	1,11	7.1616	**0.0232**
		27	69.4 ± 6.8	85 ± 12.6	1,11	1.2744	0.2853
		28	76.8 ± 7.5	65.0 ± 25.0	1,9	0.3988	0.5453
		29	87.5 ± 5.3	50.0	1,6	7.1962	**0.0437**
		30	93.0 ± 5.8	100.0	1,5	0.2402	0.6497
		31	95.0 ± 0.0	-	-	-	-
		32	100.0	-	-	-	-

Significant differences between treatments are shown in bold.

## Data Availability

The data presented in this study are available on request from the corresponding author.

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
