# Peer review of "Earwig Releases Provide Accumulative Biological Control of the Woolly Apple Aphid over the Years"

_insects, 2023, doi:10.3390/insects14110890_

Round 1
Reviewer 1 Report
Comments and Suggestions for Authors
This manuscript deals with an interesting topic, its structure is fine and conclusions are supported by the experimental work. I have only two remarks: 1) it is not clear to me if authors assessed natural earwig densities before to release them; 2) it seems that ANOVA was applied without a repeated measures approach; the latter is appropriate when sampling is done across the growing seasons.
Reviewer 2 Report
Comments and Suggestions for Authors
Modern agriculture is in constant need of technologies increasing efficacy and decreasing adverse side-effects of pest control strategies and technics. The release of entomophagous agents is among the most safe and ecology-friendly approaches as it utilizes natural mechanisms of population dynamics regulation. This is therefore expectable to increase exploitation of this component of pest management programs in sustainable agriculture, including the organic farming. Certain groups of natural enemies are widely applied to control insect pests while other remain neglected or understudied, and this seems to the case of the earwigs. The paper under review deals with the use of earwigs against aphids for the sake of orchard protection, and therefore it perfectly suits the journal scope.
My suggestions are somewhat personal as I’m a specialist in insect pathology and parasite-host interactions, preferring to review papers on molecular phylogeny and intracellular parasitism, but insect predators are within my scope of scientific interests while apple orchard is my family business. Anyway, I will do my best to judge from the scientific point of view.
It seems that the amount of published papers dedicated to this problem do not cover it in full and further studies are essential for a more comprehensive understanding of natural predator-prey interactions in agricultural landscapes and possible ways to use it for the needs of crop production. In particular, previous studies concerned either single release or repetitive releases within one year. The current work offers “single release” (as the authors call it) repeated over three years. And though this method may not be fully esteemed by those producers dreaming of “quick money” when direct investment results in immediate profit, from the scientific point of view, steep year-by-year development provides accumulative effect with a long-term outcome. After all, an apple orchard is an enterprise which exists for decades, if not for centuries, and continuous amelioration of its health serves for the generations of users. These reasons are behind my positive opinion towards the study. The methods and results are nicely described, the illustrations are comprehensive, and the conclusions are logical and convincing. The only two variants, “earwig release” and “control”, are good enough to initiate these studies. Further research might include different loads of released predators and other factors, but here we already have a sound base in the beginning.
I vote for minor revision, as the research is sound, the manuscript is well-structured and nicely written. Few comments and corrections are below.
1) It is expected to provide more detailed data concerning the pesticides applied, especially given that you deal with an organic farming system. Please bear in mind that the “pesticide” term is understood differently across the world and in some countries, the “organic farming” is not compatible with the “pesticide” use (whatever it means)
2) I wouldn’t call it a “single release” as you do it repeatedly, even though it’s not within a single season
3) I would suggest a visual representation of aphid colony length. It is essential to understand why only length, and not the width, density and other characters of the colonies were estimated
4) Watch out inconsistent grammar forms, such as switching between different styles of word endings, e.g. “randomised” vs “randomized”
5) What do you mean by “pests must be managed … to maintain .. pest population for the next season” (Lines 51-53)? Is it really so that you don’t want to eliminate the pest completely but rather plan to maintain it for some reasons? If so, please elaborate this paragraph to convince the reader
6) What do you mean by the acronym “SCI”? (Line 177)
7) Please make it clear, does “one tree” correspond to one replicate or what is meant by plural “replicates” (Line 107)
8) Please explain, how “infestation level” was estimated and how the trees with “similar infestation levels” were chosen (Lines 120-122)
9) In Figure Legends, it is unclear what states for “ns”. It would also be nice if the same color scheme was applied for different parts of one figure, for example, green line for year 2019 both in Figs. 3a & 3b
10) When you discuss the effect of earwig release “on biological control”, it becomes a logical mess as you separate “release” and “control” as totally different things and talk about one making an effect on another (Lines 201-202), though “release” is a method of “control”. You may need to rephrase this sentence
